# Viruses Ubiquity and Diversity in Atacama Desert Endolithic Communities

**DOI:** 10.3390/v14091983

**Published:** 2022-09-07

**Authors:** Leora Busse, Mike Tisza, Jocelyne DiRuggiero

**Affiliations:** 1Department of Biology, Johns Hopkins University, Baltimore, MD 21218, USA; 2The Alkek Center for Metagenomics and Microbiome Research, Department of Molecular Virology and Microbiology, Baylor College of Medicine, Houston, TX 77030, USA; 3Department of Earth and Planetary Sciences, Johns Hopkins University, Baltimore, MD 21218, USA

**Keywords:** metavirome, extreme environment, dryland ecosystems, endoliths

## Abstract

Viruses are key players in the environment, and recent metagenomic studies have revealed their diversity and genetic complexity. Despite progress in understanding the ecology of viruses in extreme environments, viruses’ dynamics and functional roles in dryland ecosystems, which cover about 45% of the Earth’s land surfaces, remain largely unexplored. This study characterizes virus sequences in the metagenomes of endolithic (within rock) microbial communities ubiquitously found in hyper-arid deserts. Taxonomic classification and network construction revealed the presence of novel and diverse viruses in communities inhabiting calcite, gypsum, and ignimbrite rocks. Viral genome maps show a high level of protein diversity within and across endolithic communities and the presence of virus-encoded auxiliary metabolic genes. Phage-host relationships were predicted by matching tRNA, CRISPR spacer, and protein sequences in the viral and microbial metagenomes. Primary producers and heterotrophic bacteria were found to be putative hosts to some viruses. Intriguingly, viral diversity was not correlated with microbial diversity across rock substrates.

## 1. Introduction

Viruses are the most abundant and genetically diverse biological entity [1,2]. Studies of the viral landscape across environments underlined the essential roles played by viruses in microbial communities, including mediating the host metabolism, conferring antibiotic resistance to hosts [3,4], and modulating the structure of communities via infection and phage-induced mortality [5]. Approximately 20% of bacteria are infected by viruses daily in the marine environment, and 10–20% undergo phage-induced lysis [5,6]. Viral metagenomic studies have identified auxiliary metabolic genes (AMGs) [7,8], proteins associated with extreme tolerance [9], and transcriptional/translational regulators within viral sequences [10]. While CRISPR/Cas systems are often found in bacteria and archaea to defend against invading viral genomes and other foreign DNAs, viruses have been found to carry CRISPR spacer arrays from their respective hosts or to encode anti-CRISPRs (Acrs) proteins [11,12].

A remarkably underexplored area of viral metagenomics is in extreme ecological zones. Viruses have been reported in hydrothermal vent plumes [13], in Siberian permafrost [14], in Antarctic soils and lakes [15,16,17,18], in soils from the Sahara, Namib, and Atacama Deserts [9,19,20,21,22], on the ventral side of quartz rocks (hypolithons) in the Namib Desert [23], and in halite nodules from the Atacama Desert [24,25]. While models proposing a symbiotic relationship between viruses from desert soils and host bacteria have been proposed [9], knowledge about viruses’ diversity and roles in desert microbial communities is severely lacking.

Deserts are unique ecological niches characterized by intense solar radiation and extreme water scarcity [26]. While desert soils retain low numbers of culturable bacteria [27,28], functionally diverse microbial communities have been documented within rock substrates (endoliths) often described as environmental refuges for life [26,29,30]. The substrate’s physical structure enhances water retention and protects against desiccation, solar radiation, wind, and temperature fluctuations [31]. Endolithic communities have been documented in hot and cold deserts worldwide and in various substrates [26,29,30]. In halite nodules, *Halothece* cyanobacteria and unique algae are the primary producers of a community dominated by heterotrophic haloarchaea [24,32]. In other substrates such as sandstone, calcite, gypsum, ignimbrite, and granite, *Chroococcidiopsis* cyanobacteria are the primary producers cohabiting with heterotrophic bacteria from the *Actinobacteria*, *Chloroflexi*, and *Proteobacteria* [33,34,35,36]. However, the presence and activity of viruses in endolithic communities remain vastly unexplored.

The Atacama Desert in Northern Chile is one of the oldest and most inhospitable environments on Earth [37,38], receiving <1 mm of rain in its hyper-arid core annually [39]. Previous work indicated that gypsum, ignimbrite, and calcite rocks from the Atacama Desert harbor functionally diverse microbial communities [34,40]. To thoroughly evaluate the viral landscape and the role of viruses in shaping the microbial community structure in the endolithic habitat, we characterized the viromes’ taxonomic composition, abundances, infective strategies, functional annotations, and phage–host relationships in endolithic substrates from the Atacama Desert.

## 2. Materials and Methods

*Substrate Collection and Metagenomic Libraries*. Calcite (n = 3) from Valle de la Luna (VL), gypsum from Cordon de Lila (CL) (n = 3), KM37 (KM) (n = 3), and Monturaqui (MTQ) (n = 3), and ignimbrite rocks from MTQ (n = 3) were previously collected, the DNA was extracted, and metagenomic libraries were sequenced, as previously described [40]. The assembled library sizes were 217 Gbp (3 libraries, 81,607 contigs) for calcite (JGI IMG Taxon OID: 3300039108, JGI Project ID: Gp0402248), 320 Gbp (9 libraries, 120,629 contigs) for gypsum (JGI IMG Taxon OID: 3300028913, Project ID: Gp0195499), and 164 Gbp (3 libraries, 50,371 contigs) for ignimbrite (JGI IMG Taxon OID: 3300039169, Project ID: Gp0402255), respectively (Appendix A).

*Viral Identification, Classification, and Functional Capacity*. Viral contigs were identified and annotated in assembled metagenomes for each substrate with VirSorter2 2.1.0 [41], with the default settings (Appendix A). Contigs > 3 kbp and containing a viral hallmark gene were retained. The end_to_end pipeline of CheckV v0.7.0 [42] was used to estimate the quality and completeness of the viral contigs, and contigs that were <5% complete were discarded from further analysis. The functional annotation of viral sequences and taxonomic classification at the family level were assigned based on the contig similarity with Cenote-Taker2 v2.1.3 [43]. Viral contig similarity and genotypic clustering were assessed with vConTACT2 v0.9.19 [44] using the NCBI Bacterial and Archaeal Viral RefSeq v94 database. Gene similarity networks were constructed at the genus level using Cytoscape v3.8.2 [45]. Singleton viruses were added to display the total count of viruses processed with Cenote-Taker2. Clinker v0.0.21 [46] was used to visualize gene cluster comparisons between viral contigs from a single substrate. Viral contigs’ breadth and depth coverage was calculated via CMSeq v1.0.4 [47]. The depth median calculated by CMSeq was used as a proxy to evaluate relative viral abundances. The default pathway of the Phage Classification Tool Set (PHACTS) [48] was employed to classify the lifestyle of all viruses. AMGs were identified using VIBRANT v1.2.1 [7] and DRAMv v1.3.2 [49].

*Computational Phage–Host Prediction*. Host prediction was performed using a suite of computational tools. Diamond [50] was used for protein–protein similarity analysis, and proteins with the highest amino acid identity between viral and bacterial sequences were used to predict hosts. The CRISPRCasTyper program [51] was used to identify and annotate CRISPR-Cas loci and spacer sequences. Viruses’ spacer sequences and CRISPR proteins were BLAST searched against the rock metagenomes to assign hosts. The search for tRNA genes in the viral contig was carried out with ARAGORN v1.2.41 [52]. Identical matches between viral and microbial tRNA sequences were used to infer putative hosts.

## 3. Results

We used previously acquired metagenomes from endolithic desert communities [34,40] to search for the presence of viral contigs. Individual rocks from three types of endolithic substrates—calcite, gypsum, and ignimbrite—were collected from several locations in the Atacama Desert in Chile (Figure 1): Metagenomic libraries—three for calcite, nine for gypsum, and three from ignimbrite, each from a different rock—were previously described (Appendix A) [34,40].

### 3.1. Endolithic Metagenomes from the Atacama Desert Harbor Diverse Viruses

Using VirSorter2, we found 2911 “putative” viral contigs over 500 nt, with 1524 contigs in the calcite, 187 in the gypsum, and 1200 in the ignimbrite metagenomes. Viral contigs represented 1.9%, 0.2%, and 2.4% of the endolithic metagenome assemblies for calcite, gypsum, and ignimbrite. Due to the short median length (1.801 kbp) and the associated false positive rate with the virus prediction of short contigs, we filtered the set of 2911 virus contigs down to contigs greater than 3 kb, encoding one or more virus “hallmark” gene (VirSorter2), and with completeness greater than 5%, as predicted by CheckV (Appendix A). After filtering and dereplication, we identified and annotated 100 unique “high confidence” viruses, defined as distinct viral genomes (Figure 1, Appendix A). Almost all viral sequences were of “low” quality (<50% complete), with only five of “medium” quality (50–90% complete) and two of “high” quality (>90% complete).

The ignimbrite metagenome had the highest number of unique viruses (53 viruses), while calcite (35 viruses) and gypsum (12 viruses) had fewer (Figure 1; Appendix A). On average, gypsum viral genomes were slightly longer (18.6 kbp) than the calcite (11.0 kbp) and ignimbrite (9.8 kbp) viral genomes (Figure 1). The average G+C contents of viruses were similar across substrates, with 61.1% for the calcite, 53.9% for the gypsum, and 64.4% for the ignimbrite. The breadth and depth of viral genomes were calculated with CMSeq for each rock substrate. The sequencing breadth was constantly ~one across viruses in the calcite and ignimbrite and was variable across the gypsum collection sites (Figure 2). The average read depth was higher in viruses from calcite (8.9–483.9) and ignimbrite (3.7–254.9) compared to those from gypsum (1–11.7). Some viruses in the gypsum overlap between the geographically dispersed CL, KM, and MTQ collection sites. Viral genomes were segregated by rock type—there was no overlap between calcite, gypsum, and ignimbrite viruses.

Most of the viruses we identified were novel viruses. Using a protein–protein similarity network analysis, we found that 70 viruses shared no protein similarity to other viruses in the data or the RefSeq prokaryotic virus references (Figure 3). Fourteen clustered only with each other, and the clusters were substrate-dependent. Only 16 had similarities with previously identified viruses in the RefSeq database. Overall, 6 of 35 viruses in calcite, 8 of 12 viruses in gypsum, and 15 of 55 viruses in ignimbrite formed clusters with each other or with RefSeq viruses. Using Cenote-Taker2, we found three viral families in the calcite and ignimbrite communities, the *Siphoviridae*, *Myoviridae*, and *Podoviridae*, and all double-stranded DNA and head-tail viruses (Figure 3). Viral genomes identified only at the level of the *Caudoviricetes* class were also found in the calcite and ignimbrite communities. In contrast, archaeal *Haloviruses* were only found in the ignimbrite community, albeit at a very low relative abundance. The gypsum community harbored only two viral families, the *Siphoviridae* and *Myoviridae*. *Siphoviridae* viruses dominated all the viromes (Appendix A).

### 3.2. Viral Genome Organization Is Highly Diverse across Endoliths

Clinker was used to generate viral genome maps based on the protein–protein amino acid identity across viruses. Most proteins shared little amino acid identity, even between viruses from the same rock type (Figure 4A). Only two clusters, formed by viruses with any level of protein identity across their genome, were found for gypsum viruses, while 11 were found for calcite viruses and 12 were found for ignimbrite viruses. Each cluster contained between two and five viruses (Appendix A). Viral structural and replication proteins shared higher levels of amino acid identity in calcite and gypsum viruses compared to other proteins. In contrast, DNA packaging proteins share higher identity levels in ignimbrite viruses. Other genes, such as those coding for AMG and CRISPR proteins, were virus-specific and did not retain amino acid similarity. Overall, the viruses shared little genomic similarity in terms of genome organization, genome size, and protein sequence identity (Figure 4B and Appendix A).

Despite the high protein diversity across all viruses, we found a small number of genome map clusters with viruses from ignimbrite together with viruses from either calcite or gypsum (Figure 4B). In these clusters, proteins sharing the most amino acid identity encoded for viral structure, replication, or DNA packaging proteins.

### 3.3. Viral Infective Strategies Vary by Viral Family

We assessed viral infective strategies using PHACTS to determine whether the endolithic viruses had a lytic (productive infection) or temperate (viral DNA integrated into host DNA or prophage formation) life cycle. There was no preference for either life cycle in the calcite and ignimbrite viromes; however, a higher proportion of gypsum viruses were associated with a lytic lifestyle (Figure 5A and Appendix A). Given the small number of viruses we identified in the gypsum, the statistical significance of the lytic preference is low. At the family level, some viruses favored one life cycle over the other; more *Podoviridae* and *Microviridae* viruses had a lysogenic lifestyle, whereas more *Siphoviridae*, *Myoviridae*, and *Halovirus* viruses had a lytic lifestyle. Viruses only identified at the *Caudoviricetes* level also favored a lysogenic lifestyle (Figure 5B).

### 3.4. Endolithic Viruses Contain a Diverse Array of Auxiliary Metabolic Genes

Viruses may enhance their replication by taking up auxiliary metabolic genes that boost their host metabolic activity (AMGs) [7,8]. AMGs are thought to be beneficial metabolic genes derived from a host DNA, which a virus integrates into its genome and which might amplify its replication. Using VIBRANT, we identified 20 AMGs in 11 endolithic viruses (Figure 6A). Genome maps of viruses containing AMGs identified and labeled by VIBRANT and DRAMv were constructed (Figure 6B). Carbohydrate, cofactor, and amino acid metabolism genes were present in viruses from multiple substrates, while other AMGs were substrate-specific. Notably, at the pathway level, sulfur metabolism, folate biosynthesis, and a sulfur relay system were present in viruses from calcite. In viruses from gypsum, we found genes related to phosphonate metabolism, porphyrin/chlorophyll metabolism, and xenobiotic degradation pathways. Lastly, genes related to glycolysis, the TCA cycle, pyruvate metabolism, amino acid, and glycan biosynthesis pathways were identified in ignimbrite viruses. When we conducted the AMGs analysis with DRAMv, we only found 13 AMGs—seven in calcite, five in gypsum, and one in ignimbrite viruses (Appendix A). Nucleotide biosynthesis pathways were found in viruses from calcite and gypsum. A methionine degradation pathway was found in viruses from ignimbrite, suggesting organic nitrogen metabolism.

### 3.5. Viral–Host Relationship Prediction

We used the ARAGORN tool to predict the tRNA sequences in endolithic viruses. We found three tRNAs in calcite viruses, one in gypsum viruses, and two in ignimbrite viruses (Table 1). We used a 100% tRNA sequence match to predict a virus’s host. BLAST searches of the endolithic metagenomes identified unclassified bacterial hosts across the three substrates, with one *Cyanobacteria* host found in the calcite community.

We identified two ignimbrite viruses containing CRISPR proteins or spacer arrays indicating phage–host interactions (Table 2, Appendix A). Five spacers and a CRISPR-associated protein were found in one ignimbrite virus (Ignimbrite_2ct1), and four consecutive CRISPR proteins were found in another ignimbrite virus (Ignimbrite_2ct2), using CRISPRCasTyper [51]. BLAST searches of viral CRISPR proteins in the endolithic metagenomes revealed potential hosts encoding similar CRISPR proteins (Table 2). *Cyanobacteria*, *Actinobacteria*, *Proteobacteria*, and unclassified bacteria metagenomic sequences indicated potential phage–host interactions with ignimbrite viruses. The virus-encoded CRISPR spacers (Appendix A) matched the CRISPR spacers on a contig from an unclassified bacterium. The CRISPR-associated endonuclease Cas12f3 from the same virus was most similar to the putative *Cyanobacteria* hosts (Table 2).

## 4. Discussion

Endolithic substrates from arid deserts harbor diverse microbial communities spanning multiple domains of life and trophic levels [29,36]. Our investigation of the metagenomes of endoliths from the Atacama Desert revealed that the viromes of calcite, gypsum, and ignimbrite rock communities were novel and highly diverse. We assembled 100 high-confidence virus genomes from these metagenomes, with only a few having an even remote similarity to viral genomes from public databases. Our analysis revealed the presence of *Siphoviridae, Myoviridae*, and *Podoviridae* viral families, one ssDNA virus belonging to the family *Microviridae* in the calcite metagenome, and archaeal *Haloviruses* identified in the ignimbrite metagenomes. This extent of the diversity of viruses in desert environments has previously been reported in desert sands [19,53,54], soils from the Atacama and Namib Deserts and the Dry Valleys of Antarctica [9,16,22], and hypolithons from the Namib Desert and Antarctica [17,23], illustrating the extent of host–virus interactions in extreme environments [55].

The taxonomic annotation of the Atacama endolithic viruses was consistent with that of previous studies; in Antarctic hypoliths, 74% of the virome was constituted by *Siphoviridae* and *Myoviridae* [17], while Namib Desert hypoliths retain viruses of the order *Caudovirales*—principally, *Siphoviridae* [23]. In Atacama Desert soils, 84 viral population genomes were identified, many of which belong to the *Siphoviridae*, *Myoviridae*, and *Podoviridae* viral families [9]. Halite nodules from the Atacama Desert harbored mostly *Haloviruses*, most likely due to the overwhelming representation of haloarchaea in the hypersaline community, in addition to bacterial (*Myoviridae, Siphoviridae,* and *Podoviridae*) and archaeal (*Pleolipoviridae* and *Sphaerolipoviridae*) viruses [24,25,56]. These previous studies support the idea that the viral composition of the calcite, gypsum, and ignimbrite communities was representative of viruses from harsh and hyper-arid drylands.

Analyzing viral infective strategies in microbial communities can provide insights into the power viruses exert on community structure and functioning. While we found no preference for a lytic or lysogenic life cycle in the calcite and ignimbrite viromes, a small increase in lytic viruses was found in the gypsum virome, albeit not statistically significant. In contrast, several studies reported the prevalence of lysogenic and pseudolysogenic viruses in warm deserts and Antarctica soils and hypolithons, suggesting that environmental stressors place selective pressure on viruses to seek refuge within host cells [9,17,19]. It is only when conditions become more favorable for the host, such as rainfall or higher atmospheric humidity, that desert soil viruses enter the lytic cycle [9]. In contrast to soil and hypolithons, most halite endolithic viruses were lytic rather than lysogenic [24]. These viruses were from halite nodules from Salar Grande, a location in the Atacama Desert with a high relative humidity [57,58]. High air relative humidity is associated with the constant deliquescence of halite nodules, as measured by in situ sensors, providing liquid water to the community within [57,59]. Furthermore, actively transcribing viruses were reported in this community, although transcriptionally active viral contigs were not correlated with viral abundance [25]. The more balanced lysogenic versus lytic viral abundances found in Atacama endolithic communities indicate that the endolithic habitat is more favorable for virus infectious cycles than desert soils.

Identifying phage–host relationships computationally presents a unique challenge, particularly when most of the viruses identified have few reference genomes in databases [23]. Using tRNA, CRISPR spacers, and protein sequence similarity, we identified eight putative phage–host interactions with *Cyanobacteria*, *Actinobacteria*, and *Proteobacteria* as potential hosts. We also identified two viruses containing either CRISPR spacers or CAS proteins. Comparatively, phage–host interactions were reported for 15 viruses of 85 viral populations identified in Atacama soils [9]. The putative hosts were *Actinobacteria*, *Chloroflexi*, and *Firmicutes*, and some viruses were found to match more than one taxonomically distant host when using a kmer frequency-based matching algorithm. Challenges were reported with Namib Desert hypolithons, where host identification was not entirely consistent with the taxonomic composition of the microbial community; unexpectedly, cyanophages were not identified, despite a community dominated by *Chroococcidiopsis* sp. [23]. A paucity of cyanophages was also reported in the virome of Antarctica hypolithons [17]. The hosts we identified in the Atacama Desert endoliths were from highly abundant taxa in the community. Among the putative hosts were *Cyanobacteria*, suggesting functional interactions between viruses and primary producers. Cyanophages were also reported in halite endoliths [24,25] and Polinton-like viruses, for which the *Dolichomastix* algae, the only eukaryotes in that system, might be the host [57]. While tRNA sequence matching revealed putative hosts for six viruses across the three endolithic substrates, we found several tRNAs in viral genomes that did not match any cellular tRNA in our metagenomes, suggesting that deeper sequencing is required to parse out taxonomic relationships between viruses and their respective hosts.

Our findings indicated that a high level of protein diversity existed among endolithic viruses, even within the same rock community. When significant amino acid similarities between viral proteins of different rock substrates were observed, it was mostly for structural, replication, and packaging proteins underlying their essential functions. We also found several viral auxiliary metabolic genes (AMGs) across the three substrate communities. AMGs are derived initially from the host, potentially providing fitness advantages during active viral infection [60]. Previous studies showed that natural environments tend to encode amino acid and cofactor/vitamin metabolism AMGs, while AMGs derived from human or urban environments tend to encode products for amino acid, cofactor/vitamin, and sulfur relay metabolism [7]. AMGs involved in extremotolerance, such as genes encoding for sporulation proteins, hydrolases, quorum sensing regulators, and a WhiB-like transcription factor, were found in the viral metagenomes of Atacama Desert soil communities. This finding led to a mutualistic model in which viruses seek refuge within protective host populations in exchange for delivering extreme tolerance genes to their hosts [9]. In contrast, a high frequency of bacterial cell wall degradation and ribonucleotide reductase genes were found in the virome of Namib desert hypoliths [23]. In our study, AMGs related to sulfur, glycan, energy, and aromatic compound metabolism were substrate-specific. At the same time, cofactor/vitamin, carbohydrate, and amino acid synthesis were found in multiple substrates, indicating the active role viruses may have in shaping endolithic microbial community dynamics.

Endolithic microbial diversity was previously found to correlate with the substrate’s architecture, defined as its physical structure and water retention capabilities [34,36,40]. Gypsum harbored the most diverse microbial community, followed by calcite and ignimbrite [34]. It was, therefore, intriguing to find that the viral diversity of these communities followed the opposite trend. Indeed, we found that the ignimbrite community had the highest number of unique viruses, whereas gypsum had the lowest. Six different viral families were present in the calcite and ignimbrite rock communities, and only two were identified in the gypsum. This apparent discrepancy might be related to the architecture of the substrates; the larger pores in the gypsum, and, to a lower extent, in the calcite, might allow for better water retention, the mixing of nutrients, and increased microbial interactions and viral infections, resulting in a more homogeneous viral population [34]. In contrast, the ignimbrite has smaller pores, which are often not connected with each other and the surface, limiting water retention and nutrient circulation [34]. The resulting microenvironment heterogeneity might provide for the survival of a more diverse assembly of hosts and viruses that might otherwise be outcompeted. While not previously applied to viruses, this relationship between spatial heterogeneity and diversity has previously been reported for soil microbial communities [61,62]. Furthermore, because Meslier’s assessment of diversity was at the phylum level, it is possible that the analysis did not represent a higher level of strain diversity in the ignimbrite community. Within the isolated pores of the ignimbrite substrate, increased heterogeneity at the strain level would allow more diverse bacterial strains to coexist, accounting for the observed higher viral diversity in ignimbrite.

## 5. Conclusions

We identified, for the first time, endolithic viruses in calcite, gypsum, and ignimbrite microbial communities from the hyper-arid Atacama Desert. Most of the 100 unique viruses we identified were novel and highly diverse. We confirm the critical role of viruses in shaping microbial communities in extreme environments via the sharing of AMGs, tRNAs, CRISPR spacers, and Cas proteins. Host identification remains a complex problem, and expanding the use of metagenomic tools and experimental approaches tailored to identify host–virus interactions in microbial communities from other deserts or along moisture gradients will provide greater insight into the role of viruses in shaping microbial communities in one of the most rapidly expanding biomes on earth—drylands.

## Figures and Tables

**Figure 1 viruses-14-01983-f001:**
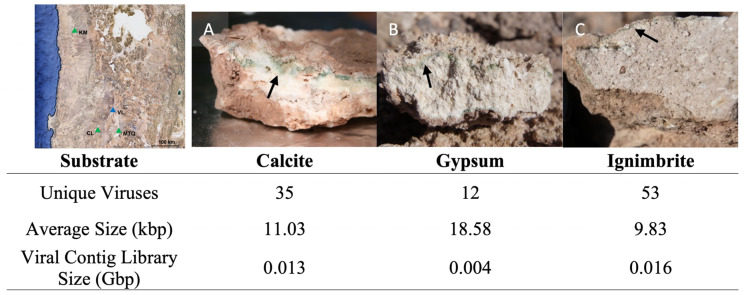
Endolithic substrates from the Atacama Desert used in this study. (Left) Map of the Atacama Desert showing sampling locations. (Right) Cross-sections of rock substrates with green colonization zones (black arrows) for (**A**) calcite from Valle de la Luna (VL), (**B**) gypsum from Cordon de Lila (CL) (gypsum was also collected from KM and MTQ), and (**C**) ignimbrite from MTQ.

**Figure 2 viruses-14-01983-f002:**
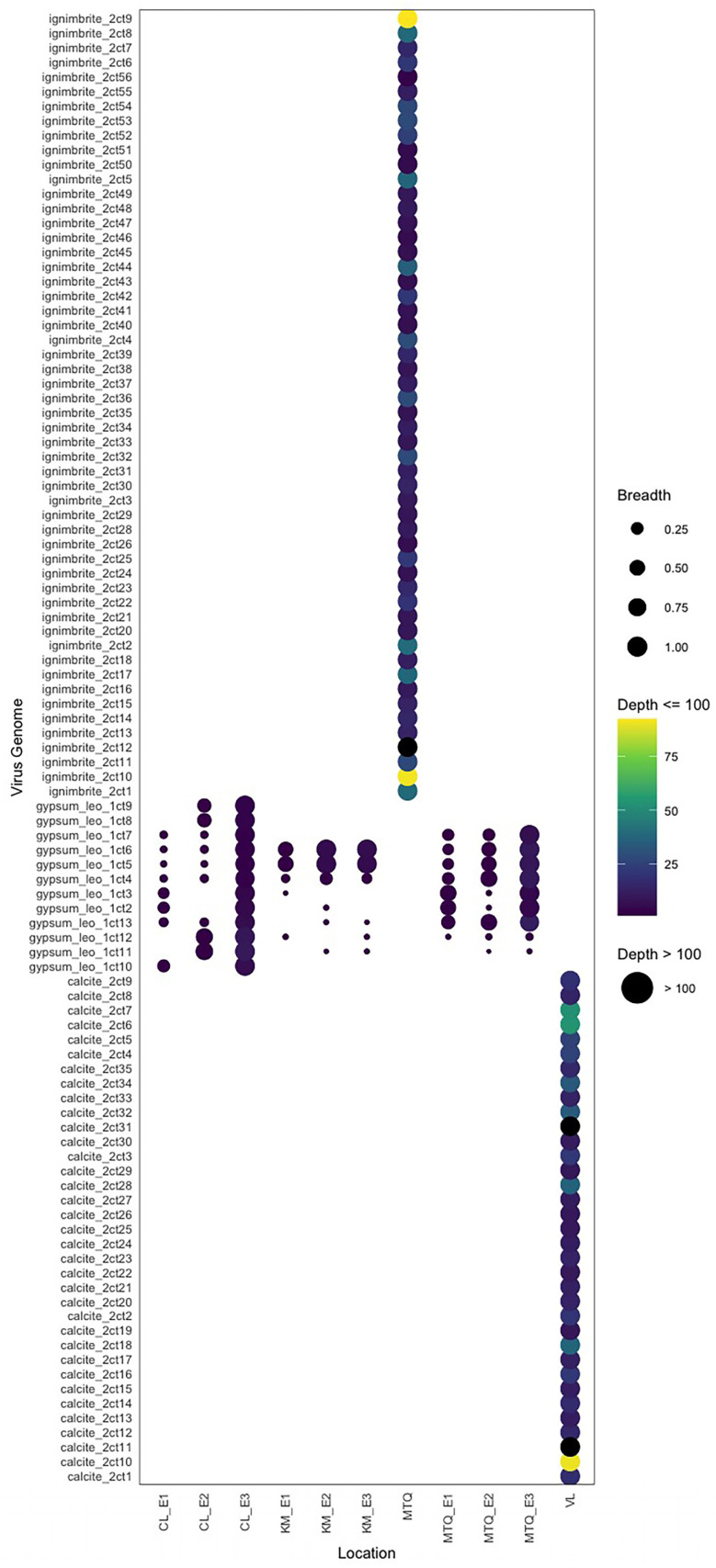
Sequence breadth and depth of viral genomes in the calcite, gypsum, and ignimbrite endolithic metagenomes.

**Figure 3 viruses-14-01983-f003:**
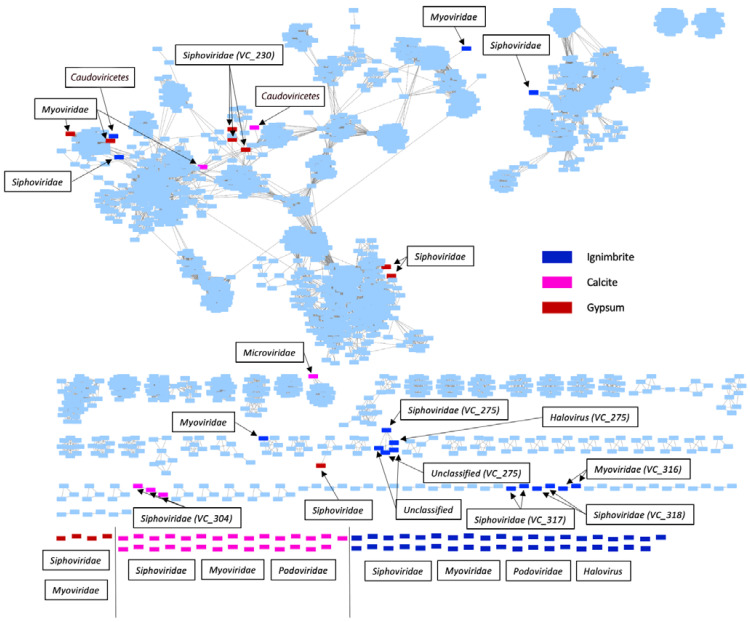
Protein similarity network for viral contigs identified in the calcite (pink), gypsum (red), and ignimbrite (navy blue) endolithic metagenomes and the RefSeq viruses (sky blue). Nodes are represented by rectangles and colored by the origin of viral contigs; edges are represented by gray lines. Cenote-Taker2 was used to identify viral families. Taxonomy at the genus level and viral clusters (VCs) were predicted by vContACT2. Diamond was used to assess protein–protein similarity. Singletons with little-to-no genomic similarity to viruses in this study or the reference database are not shown.

**Figure 4 viruses-14-01983-f004:**
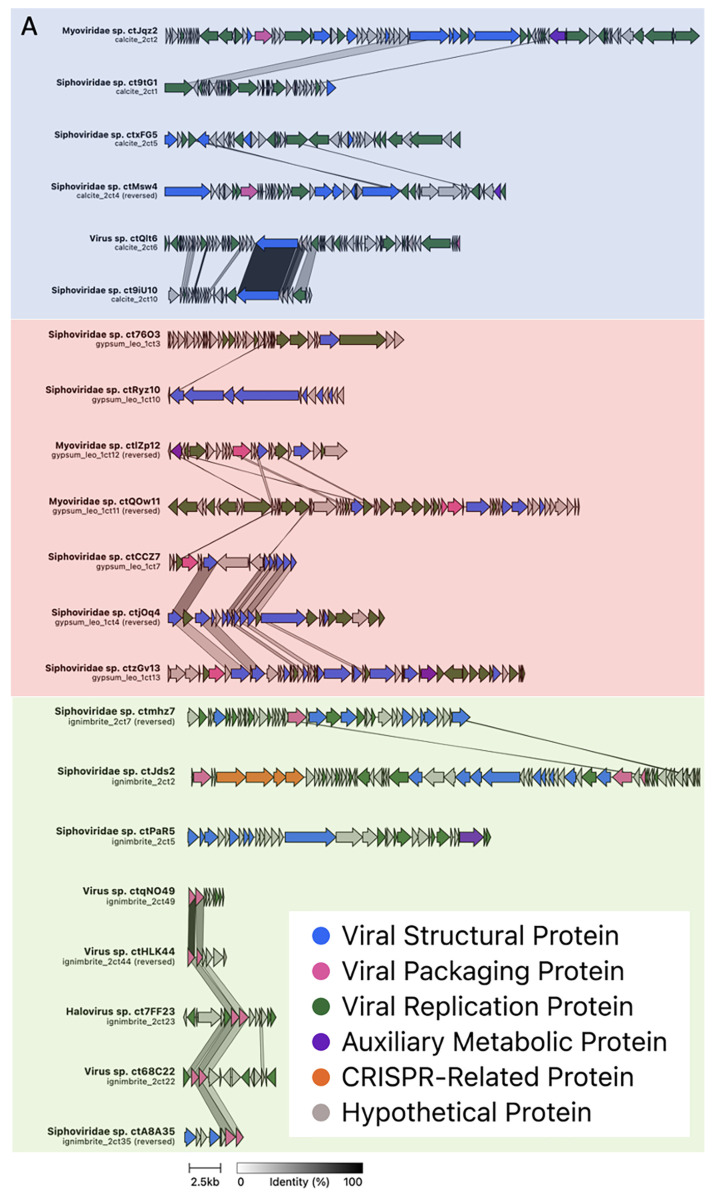
Genome maps based on the protein similarity of viruses from calcite, gypsum, and ignimbrite endolithic communities. Protein functions are color-coded, and amino acid identity is indicated by the grayscale. Maps were drawn with Clinker. (**A**) Viral clusters in calcite (green), gypsum (pink), and ignimbrite (blue). (**B**) Viral clusters across substrates.

**Figure 5 viruses-14-01983-f005:**
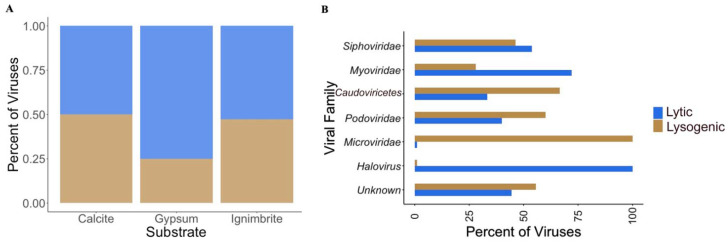
Life cycles in viruses from endolithic metagenomes. Abundances of lytic and lysogenic viruses (**A**) for each substrate and (**B**) per viral class or family across all substrates; viruses in the *Caudoviricetes* were only identified at the class level.

**Figure 6 viruses-14-01983-f006:**
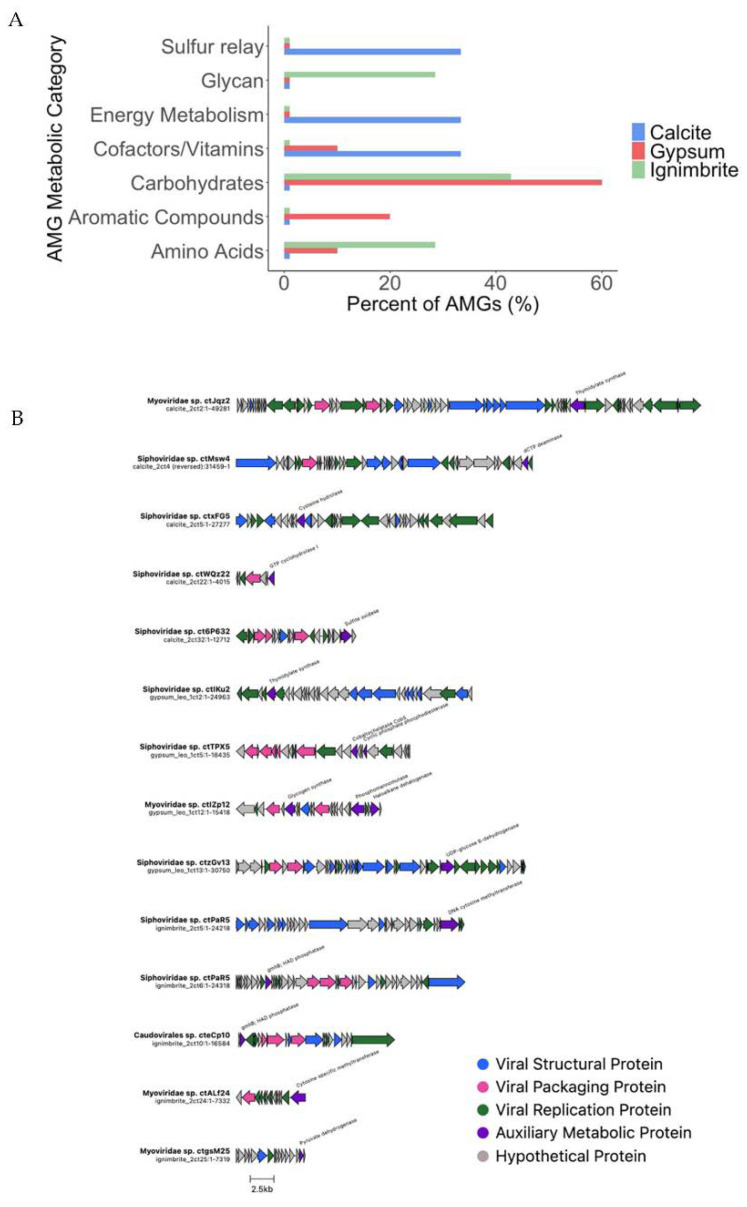
Auxiliary metabolic genes (AMGs) in endolithic viral genomes. (**A**) Relative frequency of AMGs categories, as determined by VIBRANT. (**B**) Genome maps of viruses containing AMGs annotated by VIBRANT and DRAMv.

**Table 1 viruses-14-01983-t001:** tRNA sequences and predicted hosts identified for calcite, gypsum, and ignimbrite viruses.

Virus-ID	tRNA Length (bp)	% GC	tRNA Identity	Putative Host ^(1)^	Number of Predicted Hosts
Gypsum_1ct11 (Myoviridae sp. ctQ0w11)	94	83.0	Alanine	Bacteria	1
Calcite_2ct6 (virus sp. ctQIt6)	76	47.4	Valine	Bacteria *Cyanobacteria*	3 1
Calcite_2ct17 (*Siphoviridae* sp. Ct19o17)	79	82.3	Alanine	Bacteria	2
Calcite_2ct32 (*Siphoviridae* sp. Ct6P632)	85	61.2	Tyrosine	Bacteria	1
Ignimbrite_2ct7 (*Siphoviridae* sp. ctmhz7)	77	70.1	Methionine	Bacteria	1
Ignimbrite_2ct8 (*Siphoviridae* sp. ctmWY8)	76	73.7	Glycine	Bacteria	1

^(1)^ Bacteria indicate a bacterial host with no lower rank taxonomy.

**Table 2 viruses-14-01983-t002:** CRISPR-associated proteins and spacer arrays and predicted hosts identified for calcite, gypsum, and ignimbrite viruses.

Virus	Viral CRISPR Protein	Putative Host ^(1)^	% Identity	e-Value
Ignimbrite_2ct1 (*Myoviridae* sp. ctz491)	CRISPR-associated endonuclease Cas12f3	Bacteria	84	5 × 10^−19^
	Spacer 1	Bacteria	100	4 × 10^−10^
	Spacer 2	Bacteria	100	4 × 10^−10^
	Spacer 3	Bacteria	100	1 × 10^−10^
	Spacer 4	Bacteria	100	5 × 10^−8^
	Spacer 5	Bacteria	100	9 × 10^−13^
Ignimbrite_2ct2 (*Siphoviridae* sp. ctJds2)	CRISPR-associated protein Csx17	Bacteria	100	0.0
		Actinobacteria	39	3 × 10^−78^
		Proteobacteria	40	3 × 10^−61^
	CRISPR-associated endonuclease/helicase Cas3	*Bacteria*	100	0.0
		*Actinobacteria*	50	0.0
	CRISPR-associated protein Csb1	*Bacteria*	100	0.0
		*Actinobacteria*	53	7 × 10^−105^
	CRISPR-associated protein Csb2	*Bacteria*	100	0.0
		*Actinobacteria*	79	6 × 10^−138^

^(1)^ Top Blast hit.

## Data Availability

The endolithic metagenomes are available from the JGI Genome Portal under the IMG taxon # 3300039108, 3300028913, and 3300039169. All analysis pipelines and visualization scripts are available in Appendix A.

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
