# Peer review of "Viruses Ubiquity and Diversity in Atacama Desert Endolithic Communities"

_viruses, 2022, doi:10.3390/v14091983_

Round 1

Reviewer 1 Report

This manuscript presents metagenomic study/comparisons of endolithic (inhabiting calcite, gypsum, and ignimbrite rocks) viromes in communities of hyper-arid deserts, providing taxonomic classification, level of protein diversity and presence of virus-encoded auxiliary metabolic genes; infective strategies, functional annotations and phage-host relationships.

The manuscript is written in a consequential and organized mode. I would just suggest adding several relevant references and making comparisons with.

The study could be evaluated as comprehensive as it could be considering the fact that the currently available methods/tools are not complete for deeper research, sequencing of metagenomes,

minimizing false-positive or false negative results, providing viral metagenomics for the exact host species to be identified and more accurate virus–host interactions’ predictions.

The manuscript is one step forward to understanding the ecology and evolution of the important group of viruses in dry-land ecosystems. Further deeper metagenomic study of the abundance and diversity of the viruses populated in the harsh environments would shed light on their interactions with hosts that shape the microbial community in hyper-arid environments. Thus, the manuscript will be interesting and useful for the readers of Viruses.

I would give some suggestions/comments to be addresses:

Line 31. “phage-induced mortality [4].”

Here should be referred:

DOI 10.1007/978-94-007-5539-0_4

Line 32. “viruses daily in the marine environment”

Here should be referred:

doi:10.1038/ismej.2017.175

Line 45. “proposed [6]”

Here should be referred:

doi: 10.3389/fmicb.2021.701414

Line 50. “often described as environmental refuges for life [23, 26].”

Here should be referred:

https://doi.org/10.1016/j.mib.2018.01.003

Line 53. “in various substrates [23, 26]”.

Here should be referred:

https://doi.org/10.1016/j.mib.2018.01.003

Line 58. “Proteobacteria [29-32]. However, the presence and activity”

Here should be referred:

https://doi.org/10.1016/j.mib.2018.01.003

Line 63. “diverse microbial communities [30, 36].”

Here should be referred:

https://doi.org/10.1016/j.mib.2018.01.003

Line 127. “number of unique viruses (53)”

the given reference of “53” is not complete. Do you mean New viruses like it is in Line 187: “Most of the viruses we identified were novel viruses”?

or endemic viruses are meant? Like here doi: 10.3389/fmicb.2021.701414 - “Bacterial and archaea that inhabit the GSL extreme environments are often endemic to the region”.

“unique viruses” Should be defined, otherwise quite often term “unique viruses” is referred to deadliest or dangerous viruses.

Lines 194-200 and whole paper. “the Siphoviridae, Myoviridae, and Podoviridae”.

New classification must be used: Siphoviruses, Myoviruses and Podoviruses

class Caudoviricetes

https://doi.org/10.3390/v13030506

Lines 195-199. “Caudovirales order were identified in the calcite and ignimbrite communities. In contrast, viruses of the Myoviridae family were only found in the calcite community … Siphoviridae and Myoviridae”;

Even based on the old classification Siphoviridae, Myoviridae, and Podoviridae is included in Caudovirales.

Line 198. “Halovirus in the ignimbrite community”.

Archaeal or bacterial Halovirus?

http://dx.doi.org/10.1016/j.mib.2015.04.001

https://doi.org/10.1371/journal.pbio.3001442

Lines 233-234. “lytic (active infection) or temperate (viral DNA integrated into host DNA) life cycle”.

Temperate refers to ability display a lysogenic cycle. Not all temperate phages can integrate their genomes into host DNA.

Would suggest to write Lytic (productive infection) or lysogenic (viral DNA integrated into host DNA or prophage formation).

Lines 235-236. “a higher proportion of gypsum viruses maintained a lytic lifestyle”.

Would suggest to write associated or referred to a lytic lifestyle rather maintained.

Lines 238-239. “Podoviridae, Microviridae, and Caudovirales viruses”

Again taxonomy needs to be changed, even according to old one Podoviridae, Siphoviridae and Myoviridae go under the Caudovirales.

Lines 239-242. “temperate lifestyle”

Suggest write lysogenic lifestyle.

Line 240. “Halovirus viruses”

Give more details if possible about Halovirus represents

http://dx.doi.org/10.1016/j.mib.2015.04.001

https://doi.org/10.1371/journal.pbio.3001442 

Lines 245-246. “Viruses may enhance their replication by taking up auxiliary metabolic genes (AMGs)”.

Here should be referred from your references “6. Hwang..” “AMGs are selected to maximize viral production by enhancing host metabolism during an infection”

Line 290. ”trophic levels [26].”

Here should be referred: https://doi.org/10.1016/j.mib.2018.01.003

Line 297: “This unexpected diversity of viruses”

Why it is unexpected?

Line 300: “illustrating the extent of host-virus interactions in extreme environments”.

Here could be referred: doi:10.1038/ismej.2017.175

Line 313. “archaea in the hypersaline community”

Here should be referred doi: 10.3389/fmicb.2021.701414

Line 320. Would replace “temperate life cycle” with lysogenic one.

Lines 322-323.pseudolysogenic

Would suggest to avoid to include pseudolysogenic, otherwise provide the details that validates the method.

From your references “6. Hwang…” -  “However, computational prediction tools tend to underestimate

lysogenic viruses (35) and cannot distinguish pseudolysogens from lytic viruses.”

Line 328. rather than lysogenic [21].

Should be referred: doi: 10.3389/fmicb.2021.701414

Lines 334-336. this could be correlated with:

From your references’ list “6. Hwang…” – “This result may also indicate that viral entities sheltered below boulders undergo lytic cycles and therefore exhibit higher abundance than microbes, while those beside boulders are primarily lysogenic”.

From your references’ list “21. Crits-Christoph..” -  suggesting that most of the viruses were lytic rather than lysogenic, in contrast to viruses found in high temperature environments (Anderson et al., 2015).

And doi: 10.3389/fmicb.2021.701414

“The nutrient abundant regions of GSL support lytic lifestyle for the viruses”.

Lines 337-338. should be referred this as well:

doi: 10.3389/fmicb.2021.701414

Line 375. “In our study, AMGs related to sulfur, glycan, energy, and aromatic compound metabolism were substrate-specific”

Would suggest to correlate with study doi: 10.3389/fmicb.2021.701414

Line 384. would suggest to replace “unique viruses”

Lines 388-389. “might allow for better water retention, mixing of nutrients, and increased microbial interactions”

Here could be referred: https://doi.org/10.1016/j.mib.2018.01.003

Line 403. define unique viruses or replace it.

Suggest to add doi to references in the list where it is available. It would be easier to work on and for the readers as well.

Author Response

Reviewer 1:

This manuscript presents metagenomic study/comparisons of endolithic (inhabiting calcite, gypsum, and ignimbrite rocks) viromes in communities of hyper-arid deserts, providing taxonomic classification, level of protein diversity and presence of virus-encoded auxiliary metabolic genes; infective strategies, functional annotations and phage-host relationships.

The manuscript is written in a consequential and organized mode. I would just suggest adding several relevant references and making comparisons with.

The study could be evaluated as comprehensive as it could be considering the fact that the currently available methods/tools are not complete for deeper research, sequencing of metagenomes, minimizing false-positive or false negative results, providing viral metagenomics for the exact host species to be identified and more accurate virus–host interactions’ predictions.

The manuscript is one step forward to understanding the ecology and evolution of the important group of viruses in dry-land ecosystems. Further deeper metagenomic study of the abundance and diversity of the viruses populated in the harsh environments would shed light on their interactions with hosts that shape the microbial community in hyper-arid environments. Thus, the manuscript will be interesting and useful for the readers of Viruses.

We appreciate the positive feedback on our study.

I would give some suggestions/comments to be addresses:

Line 31. “phage-induced mortality [4].”

Here should be referred:

DOI 10.1007/978-94-007-5539-0_4

Added.

Line 32. “viruses daily in the marine environment”

Here should be referred:

doi:10.1038/ismej.2017.175

We are unsure how this article entitled “Characterization of ecologically diverse viruses infecting co-occurring strains of cosmopolitan hyperhalophilic Bacteroidetes” is relevant to our comment on viruses in the marine environment. Instead, we added

DOI:10.1126/science.abm5847

Line 45. “proposed [6]”

Here should be referred:

doi: 10.3389/fmicb.2021.701414

We are unsure how this article entitled “Viruses and Their Interactions With Bacteria and Archaea of Hypersaline Great Salt Lake” is relevant to proteins associated with extreme tolerance.

Line 50. “often described as environmental refuges for life [23, 26].”

Here should be referred:

https://doi.org/10.1016/j.mib.2018.01.003

Added.

Line 53. “in various substrates [23, 26]”.

Here should be referred:

https://doi.org/10.1016/j.mib.2018.01.003

Added.

Line 58. “Proteobacteria [29-32]. However, the presence and activity”

Here should be referred:

https://doi.org/10.1016/j.mib.2018.01.003

https://doi.org/10.1016/j.mib.2018.01.003 is a review paper that we previously cited in Lines 50 and 53; citations [29-32] are from experimental papers, which we think are more appropriate at this point of the manuscript.

Line 63. “diverse microbial communities [30, 36].”

Here should be referred:

https://doi.org/10.1016/j.mib.2018.01.003

Same comment as above.

Line 127. “number of unique viruses (53)”

the given reference of “53” is not complete. Do you mean New viruses like it is in Line 187: “Most of the viruses we identified were novel viruses”?

or endemic viruses are meant? Like here doi: 10.3389/fmicb.2021.701414 - “Bacterial and archaea that inhabit the GSL extreme environments are often endemic to the region”.

“unique viruses” Should be defined, otherwise quite often term “unique viruses” is referred to deadliest or dangerous viruses.

We are sorry for the misunderstanding; (53) is not a reference but the number of unique viruses found in the ignimbrite metagenome. Same for (35) and (12) in Line 128. We corrected this by adding “viruses” in the parentheses. The term “unique viruses” was previously defined in the result section (see also Figs. 1 and 2; Table S2).

Lines 194-200 and whole paper. “the Siphoviridae, Myoviridae, and Podoviridae”.

New classification must be used: Siphoviruses, Myoviruses and Podoviruses

class Caudoviricetes

https://doi.org/10.3390/v13030506

The paper cited above is a proposal by Turner et al. (2021) to change some aspects of virus taxonomy. While abolishing the order Caudivirales and replacing it with the class Caudoviricetes has been ratified by ICTV (Walker et al., 2022; https://doi.org/10.1007/s00705-022-05516-5), the name changes for Siphoviridae, Myoviridae, and Podoviridae have not. Therefore following the ICTV ruling, we changed Caudivirales to Caudoviricetes in the text and figures but kept Siphoviridae, Myoviridae, and Podoviridae as it was (see also the NCBI website: https://www.ncbi.nlm.nih.gov/Taxonomy/Browser/wwwtax.cgi?mode=Info&id=10662&lvl=3&lin=f&keep=1&srchmode=1&unlock#note1; https://www.ncbi.nlm.nih.gov/Taxonomy/Browser/wwwtax.cgi?mode=Info&id=10744&lvl=3&lin=f&keep=1&srchmode=1&unlock; and https://www.ncbi.nlm.nih.gov/Taxonomy/Browser/wwwtax.cgi?mode=Info&id=10699&lvl=3&lin=f&keep=1&srchmode=1&unlock)

Lines 195-199. “Caudovirales order were identified in the calcite and ignimbrite communities. In contrast, viruses of the Myoviridae family were only found in the calcite community … Siphoviridae and Myoviridae”;

Even based on the old classification Siphoviridae, Myoviridae, and Podoviridae is included in Caudovirales.

We agree that this sentence was unclear; it was replaced by “Viral genomes identified only at the level of the Caudoviricetes class were also found in the calcite and ignimbrite communities.”

Line 198. “Halovirus in the ignimbrite community”.

Archaeal or bacterial Halovirus?

We meant archaeal Halovirus; this distinction was added to the manuscript.

http://dx.doi.org/10.1016/j.mib.2015.04.001

https://doi.org/10.1371/journal.pbio.3001442

Lines 233-234. “lytic (active infection) or temperate (viral DNA integrated into host DNA) life cycle”.

Temperate refers to ability display a lysogenic cycle. Not all temperate phages can integrate their genomes into host DNA.

Would suggest to write Lytic (productive infection) or lysogenic (viral DNA integrated into host DNA or prophage formation).

This is a very good point and the changes were made in the manuscript.

Lines 235-236. “a higher proportion of gypsum viruses maintained a lytic lifestyle”.

Would suggest to write associated or referred to a lytic lifestyle rather maintained.

Changed to “were associated with a lytic lifestyle”

Lines 238-239. “Podoviridae, Microviridae, and Caudovirales viruses”

Again taxonomy needs to be changed, even according to old one Podoviridae, Siphoviridae and Myoviridae go under the Caudovirales.

See our comment above regarding taxonomy names. To clarify, we changed the sentence to “Viruses only identified at the Caudoviricetes level also favored a lysogenic lifestyle”.

Lines 239-242. “temperate lifestyle”

Suggest write lysogenic lifestyle.

Done.

Line 240. “Halovirus viruses”

Give more details if possible about Halovirus represents

http://dx.doi.org/10.1016/j.mib.2015.04.001

https://doi.org/10.1371/journal.pbio.3001442 

Unfortunately, we do not have any more details about Halovirus viruses

Lines 245-246. “Viruses may enhance their replication by taking up auxiliary metabolic genes (AMGs)”.

Here should be referred from your references “6. Hwang..” “AMGs are selected to maximize viral production by enhancing host metabolism during an infection”

We changed the sentence to “Viruses may enhance their replication by taking up auxiliary metabolic genes that boost their host metabolic activity (AMGs)” and added references.

Line 290. ”trophic levels [26].”

Here should be referred: https://doi.org/10.1016/j.mib.2018.01.003

Added

Line 297: “This unexpected diversity of viruses”

Why it is unexpected?

Changed to “This extent in the diversity of viruses in desert environments has previously been reported…”

Line 300: “illustrating the extent of host-virus interactions in extreme environments”.

Here could be referred: doi:10.1038/ismej.2017.175

Added.

Line 313. “archaea in the hypersaline community”

Here should be referred doi: 10.3389/fmicb.2021.701414

Added.

Line 320. Would replace “temperate life cycle” with lysogenic one.

Done.

Lines 322-323. “pseudolysogenic”

Would suggest to avoid to include pseudolysogenic, otherwise provide the details that validates the method.

From your references “6. Hwang…” -  “However, computational prediction tools tend to underestimate lysogenic viruses (35) and cannot distinguish pseudolysogens from lytic viruses.”

We agree that it is not clear how the studies we cited distinguished pseudolysogenic from lytic viruses but we want to be accurate in our citation. Furthermore, the terms “pseudolysogens/ pseudolysogenic” are found seven times in Hwang et al., despite their claim that “computational prediction tools tend to underestimate lysogenic viruses (35) and cannot distinguish pseudolysogens from lytic viruses.”

Line 328. rather than lysogenic [21].

Should be referred: doi: 10.3389/fmicb.2021.701414

In this sentence, we refer specifically to halite endolithic virus and therefore ref [21] is accurate. There are no halite nodules in the Great Salt Lake, which is the environment investigated in reference doi: 10.3389/fmicb.2021.701414.

Lines 334-336. this could be correlated with:

From your references’ list “6. Hwang…” – “This result may also indicate that viral entities sheltered below boulders undergo lytic cycles and therefore exhibit higher abundance than microbes, while those beside boulders are primarily lysogenic”.

From your references’ list “21. Crits-Christoph..” -  suggesting that most of the viruses were lytic rather than lysogenic, in contrast to viruses found in high temperature environments (Anderson et al., 2015).

And doi: 10.3389/fmicb.2021.701414

“The nutrient abundant regions of GSL support lytic lifestyle for the viruses”.

Thank you for those suggestions. It is not clear, however, in Hwang et al. whether the differences in the abundances of viruses versus microorganisms were the result of viral sensitivity to irradiation beside the boulders or increased lytic cycles under the boulders. We therefore rather not add this comment to our discussion. Below is the relevant paragraph in Hwang et al.

Additionally, we observed statistically significantly lower abundance of viral entities

relative to microbes (Fig. 5S) in C samples, suggesting that viruses are vulnerable to irradiation in desert environments, perhaps more so than microbes. This result may also indicate that viral entities sheltered below boulders undergo lytic cycles and therefore exhibit higher abundance than microbes, while those beside boulders are primarily (pseudo)lysogenic. Further work is required to reveal the predominant lifestyle and viability of viruses in the Atacama hyperarid core.” We therefore rather not add this comment to our discussion.

Lines 337-338. should be referred this as well:

doi: 10.3389/fmicb.2021.701414

There are many papers we could cite to support the statement that “Identifying phage-host relationships computationally presents a unique challenge” and decided to use one that is more closely related to the environment investigated in our manuscript [20].

Line 375. “In our study, AMGs related to sulfur, glycan, energy, and aromatic compound metabolism were substrate-specific”

Would suggest to correlate with study doi: 10.3389/fmicb.2021.701414

Thank you for the suggestion, but the study in doi: 10.3389/fmicb.2021.701414 does not address substrate specificity, which is the point of the sentence in Line 375.

Line 384. would suggest to replace “unique viruses”

“The term “Unique viruses” was defined in the result sections: “After filtering and dereplication, we identified and annotated 100 unique "high confidence" viruses, defined as distinct viral sequences (Fig. 1, Table S2)”.

Lines 388-389. “might allow for better water retention, mixing of nutrients, and increased microbial interactions”

Here could be referred: https://doi.org/10.1016/j.mib.2018.01.003

We think that it is more appropriate to cite here a research paper rather than a review.

Line 403. define unique viruses or replace it.

“The term “Unique viruses” was defined in the result sections – see above.

Suggest to add doi to references in the list where it is available. It would be easier to work on and for the readers as well.

We followed the reference format specified for this journal and it does not include doi.

Reviewer 2 Report

this paper is very interesting, I just list four comments for authors consideration.

1.      Line 21, what is meaning of “primary producers” in here? Is it referring to cyanobacteria?

2.      About the figure S2, since the podoviridae was found in this study (Lines 193-195), why there is no data of podoviridae in Fig. S2?

3.      About Figure 5B, Caudovirales is the order level of viral taxonomy, it includes Myoviridae, Siphovoridae and Podovidiae. Therefore, please consider the inclusion relationship among them, and redraw this figure.

4.      Table 1, suggest to change “virus” to “virus-ID”.

Author Response

Reviewer 2:

this paper is very interesting, I just list four comments for authors consideration.

  1. Line 21, what is meaning of “primary producers” in here? Is it referring to cyanobacteria?

Yes, we meant Cyanobacteria and it is explained in the 3rd paragraph of the introduction: “In other substrates such as sandstone, calcite, gypsum, ignimbrite, and granite, Chroococcidiopsis cyanobacteria are the primary producers cohabiting with heterotrophic bacteria from the Actinobacteria, Chloroflexi, and Proteobacteria”.

  1. About the figure S2, since the podoviridae was found in this study (Lines 193-195), why there is no data of podoviridae in Fig. S2?

Figure S2 represents viruses found across all sampling sites for gypsum only; the gypsum community harbored only two viral families, the Siphoviridae and Myoviridae;

  1. About Figure 5B, Caudovirales is the order level of viral taxonomy, it includes Myoviridae, Siphovoridae and Podovidiae. Therefore, please consider the inclusion relationship among them, and redraw this figure.

The Caudoviricetes (new name) viruses in Fig. 5B were only identified at the class level whereas other viruses could be identified at the family level. We added a comment in the legend to clarify this apparent discrepancy.”

  1. Table 1, suggest to change “virus” to “virus-ID”.

Done.